# MCMCINLA Estimation of Missing Data and Its Application to Public Health Development in China in the Post-Epidemic Era

**DOI:** 10.3390/e24070916

**Published:** 2022-06-30

**Authors:** Jiaqi Teng, Shuzhen Ding, Xiaoping Shi, Huiguo Zhang, Xijian Hu

**Affiliations:** College of Mathematics and System Science, Xinjiang University, Urumqi 830046, China; tengjq111@stu.xju.edu.cn (J.T.); dszspur@xju.edu.cn (S.D.); shixiaoping@xju.edu.cn (X.S.); zhanghg@xju.edu.cn (H.Z.)

**Keywords:** missing data, spatial lag model, MCMC, INLA, public health

## Abstract

Medical data are often missing during epidemiological surveys and clinical trials. In this paper, we propose the MCMCINLA estimation method to account for missing data. We introduce a new latent class into the spatial lag model (SLM) and use a conditional autoregressive specification (CAR) spatial model-based approach to impute missing values, making the model fit into the integrated nested Laplace approximation (INLA) framework. Combining the advantages of both the Markov chain Monte Carlo (MCMC) and INLA frameworks, the MCMCINLA algorithm is used to implement imputation of the missing data and fit the model to derive estimates of the parameters from the posterior margins. Finally, the economic data and the hemorrhagic fever with renal syndrome (HFRS) disease data of mainland China from 2016–2018 are used as examples to explore the development of public health in China in the post-epidemic era. The results show that compared with expectation maximization (EM) and full information maximum likelihood estimation (FIML), the predicted values of the missing data obtained using our method are closer to the true values, and the spatial distribution of HFRS in China can be inferred from the imputation results with a southern-heavy and northern-light distribution. It can provide some references for the development of public health in China in the post-epidemic era.

## 1. Introduction

Missing data are common and unavoidable in daily life. For example, in engineering design, equipment failure may cause some data to fail to be collected normally, and during market research, there may also be situations where respondents refuse to answer relevant questions. Besides, missing data often occurs in the medical context. For instance, in epidemiological surveys, equipment limitations prevent access to complete information, and in medical databases, not all patients’ clinical test results are available at a given time, leaving a portion of the attribute values vacant. In addition, data can be lost due to the failure of storage media and transmission media.

Information theory is an important discipline based on the methods of probability theory and mathematical statistics for the study of information entropy, data processing, and data transmission [1]. It has a wide range of applications in the effective processing and reliable transmission of information. In the era of big data, the data processing of complex information becomes an important part of research and analysis. Dealing with missing data has always been the first problem that researchers need to solve before conducting statistical analysis, as improper handling can lead to deviations in statistical inferences and even affect the final decision. Little and Rubin (2002) [2] gave a detailed and systematic introduction to the different missing mechanisms and imputation models of missing data. Buuren (2011) [3] supplemented and improved it on this basis and provided a demonstration of R code to make the imputation process more intuitive. At present, scholars have proposed a variety of imputation methods for the missing data problem, which can basically be divided into two categories: statistical methods and machine learning methods. Statistical methods mostly make assumptions based on the dataset itself, and then use the original dataset to impute the missing data accordingly. Common methods include expectation maximization (EM) imputation [4], regression imputation [5], and multiple imputation [6]. Machine learning methods generally impute the missing dataset by clustering with K-nearest neighbor imputation [7] and K-means imputation [8], and Bayesian networks [9] are represented. In recent years, with the rise of the machine learning boom, Bayesian networks have become a frequently used method to deal with missing data. Mason (2009) [10] proposed the use of a Bayesian method to model nonrandom missing data through the Bayesian missing imputation framework to adjust the missing covariates in longitudinal studies. Erler et al. (2016) [11] suggested that missing values can be imputed in a joint estimation framework using Bayesian methods. Zhang et al. (2017) [12] proposed a missing data processing method based on plain Bayesian and EM algorithms for the software workload problem. Ding (2020) [13] conducted a comparative study of the missing data imputation problem in normal models using Bayesian and jackknife multiple imputation methods, respectively, and concluded that Bayesian imputation results are more accurate.

The Markov chain Monte Carlo (MCMC) algorithm has now become a standard method for parameter estimation in many models. Doğan and Taspinar (2018) [14] (hereinafter DT) performed parameter estimation for the spatial error sample selection model with nonrandom missing data using MCMC; Hajime Seya et al. (2020) [15] improved on the work done by DT, and in parallel, they proposed that MCMC can handle the parameter estimation problem of the spatial lag sample selection model with nonrandom missing data. Although the MCMC can solve Bayesian inference excellently, MCMC may be limited by the speed of convergence and numerical stability when faced with larger models or more data. To address this problem, Rue et al. (2009) [16] proposed an algorithm that combines Laplace approximation with modernized numerical integration under a Bayesian framework—the integrated nested Laplace approximation (INLA)—which can significantly reduce computation time while guaranteeing the accuracy of an MCMC estimation. G’omez-Rubio et al. (2017) [17] described the realization of a new class of latent model in INLA which can be used directly for fitting spatial econometric models, and G´omez-Rubio and Rue (2018) [18] created a new approach combining INLA and MCMC, namely, MCMCINLA, and used it to fit spatial econometric models, linear regression models with missing data in covariates, Bayesian Lasso models, and mixed models. Gomez-Rubio et al. (2019) [19] also redefined the problem of missing values in regression models covariates by latent Gaussian Markov random field (GMRF) for analysis and imputation of missing data, and they applied it to the spatial model and the multiple linear regression model to overcome the problem wherein INLA cannot handle a model with missing values in covariates.

This paper proposes a new MCMCINLA imputation method for missing data and uses the hemorrhagic fever with renal syndrome (HFRS) disease data with random missing in covariates to establish a spatial lag (SLM) latent model to explore the developmental inputs to public health in China before the COVID-19 outbreak, and to provide reference suggestions for the national financial inputs to public health in the post-epidemic era. In addition, the use of the imputation effects of EM, Full Information Maximum Likelihood estimation (FIML), and MCMCINLA method on the missing data are compared to illustrate the effectiveness of the imputation method proposed in this paper.

The paper is structured as follows: Section 2 reviews the different missing mechanisms for the missing data and introduces the SLM latent model with random missing data in covariates; Section 3 gives the proof procedure of the model GMRF structure and describes the process of implementing the MCMCINLA algorithm for the SLM latent model with random missing data in covariates; Section 4 conducts numerical simulations for the model and algorithm proposed in this paper to verify the correctness of the method; Section 5 presents an empirical analysis of the reform adjustment problem of public health development in China in the post-epidemic era as an example; and finally, the conclusions and discussions are given in Section 6.

## 2. Model Building

### 2.1. Missing Mechanism

The relationship between missing data and complete data is known as the missing mechanism, which is broadly classified into three types: missing completely at random (MCAR), missing at random (MAR), and not missing at random (NMAR). MCAR occurs when the missing data is random and unrelated to both observed and unobserved data, MAR occurs when the missing data is only related to observed data, and NMAR occurs when the missing data is related to both observed and unobserved data [20].

Furthermore, missing data can be classified into four categories:$\left(Y_{obs},X_{obs}\right),\;\left(Y_{mis},X_{obs}\right),$ $\left(Y_{obs},X_{mis}\right)$, and $\left(Y_{mis},X_{mis}\right)$, depending on where the missing values are located. For the first case, when neither X nor Y contains missing data, the fit can be performed directly using INLA or MCMCINLA (see, for example, Gómez-Rubio, V. et al. (2017, 2018) [17,18]); for the second case, when the missing data are in the response variables, INLA can use its own properties to predict the missing values by directly calculating the predicted distribution of all the missing data in the response variable (see, for example, Gómez-Rubio, V. et al. (2017) [17]); for the third and fourth case, when the missing data are in the covariates, it is necessary to first define the imputation submodel as latent effects with a GMRF structure to make it suitable for the INLA framework and include the imputation in the main model, and then further perform the model fitting with the help of the INLA, which is also the focus of this paper.

### 2.2. The SLM Latent Model

In spatial statistics, the SLM has received increasingly wide attention from many scholars. It is mainly used to study the impact of the behavior of adjacent regions on the behavior of other regions of the whole system, expressed formally as:$$Y=\rho_{Lag}WY+X\beta +e,e\sim MVN\left(0,\sigma^{2}I_{n}\right),$$
where Y represents the observation vectors of *n* different regions, $\rho_{Lag}$ is the spatial autocorrelation parameter, W is the adjacency matrix, β is the coefficient vector of covariates, and the error term e obeys a multivariate normal distribution with the mean 0 and diagonal variance-covariance matrix $\sigma^{2}I_{n}$. We can also shift the term for Y and rewrite the model as:(1)$$Y={\left(I_{n}-\rho_{Lag}W\right)}^{-1}\left(X\beta +e\right),e\sim MVN\left(0,\sigma^{2}I_{n}\right).$$

The key to enabling the SLM to be implemented under the MCMCINLA is whether the model can be implemented within INLA. Normally, the SLM cannot be fitted directly with INLA, and we need to construct latent classes and redefine the original SLM as a model with GMRF so as to conform to the INLA framework. We can construct a latent class for the SLM as follows:(2)$$x={\left(I_{n}-\rho W\right)}^{-1}\left(X\beta +e\right),$$
where x denotes the vector of *n* random effects, ρ is the spatial autocorrelation parameter, W is the weight matrix, $X=\left(X_{mis},X_{obs}\right)$ are the covariates with random missing data, β is the coefficient vector of covariates, and the error term e obeys an independent Gaussian distribution with the mean 0 and precision matrix $\tau I_{n}$.

Using the constructed latent class of Equation (2) to rewrite the SLM model, we can obtain the SLM latent model:(3)$$Y=x+\varepsilon ={\left(I_{n}-\rho W\right)}^{-1}\left(X\beta +e\right)+\varepsilon ,$$
where ε is a small error term that is used to fit the model.

## 3. Algorithm Description

The INLA algorithm mainly targets models with structured additive regression models with a latent random field of GMRF. The premise of using the MCMCINLA fitting model is that the model needs to conform to the INLA framework, i.e., it has a GMRF structure in order to be solved. Therefore, this section first explains the GMRF structure of the main model and the imputation model, and then describes the implementation process of the MCMCINLA algorithm.

### 3.1. Proof of GMRF Structure

#### 3.1.1. GMRF Structure of the Main Model

If we assign a Gaussian prior with a zero mean and the precision matrix *Q* to β in Equation (2), the e obeys a Gaussian distribution with a zero mean and the precision matrix $\tau I_{n}$, with τ as a precision parameter. Then, INLA will want to obtain the joint distribution π(x,β) of x and β. By Bayes’ theorem, we have:π(x,β)=π(x|β)π(β),
and, by definition,
E(β)=0,
Prec(β)=Q,
$$J=E\left(x|\beta \right)={\left(I_{n}-\rho W\right)}^{-1}X\beta ,$$
$$var\left(x|\beta \right)=var\left[{\left(I_{n}-\rho W\right)}^{-1}X\beta +{\left(I_{n}-\rho W\right)}^{-1}e|\beta \right]\;={\left(I_{n}-\rho W\right)}^{-1}var\left(e|\beta \right){\left({\left(I_{n}-\rho W\right)}^{-1}\right)}^{\prime }\;={\left(I_{n}-\rho W\right)}^{-1}\frac{1}{\tau }I_{n}{\left({\left(I_{n}-\rho W\right)}^{-1}\right)}^{\prime }\;=\frac{1}{\tau }{\left(I_{n}-\rho W\right)}^{-1}{\left(I_{n}-\rho W^{\prime }\right)}^{-1},\;\mathrm{and}$$
$$K=Prec\left(x|\beta \right)=\frac{1}{var(x|\beta )}=\tau \left(I_{n}-\rho W^{\prime }\right)\left(I_{n}-\rho W\right).$$

Thus,
(4)$$\pi \left(x,\beta \right)=\pi \left(x|\beta \right)\pi \left(\beta \right)\;\propto \exp\left\{-\frac{1}{2}{\left(x-J\right)}^{\prime }K\left(x-J\right)\right\}\exp\left\{-\frac{1}{2}\beta^{\prime }Q\beta \right\}\;=\exp\left\{-\frac{1}{2}\left(x^{\prime }Kx-x^{\prime }KJ-J^{\prime }Kx+J^{\prime }KJ+\beta^{\prime }Q\beta \right)\right\}\;=\exp\left\{-\frac{1}{2}{\left(x,\beta \right)}^{\prime }P\left(x,\beta \right)\right\},$$
where P is the precision matrix of (x,β) with the structure:(5)$$\begin{array}{ccc} P & = & \left(\begin{array}{cc} K & -K{\left(I_{n}-\rho W\right)}^{-1}X\\ -X^{\prime }{\left(I_{n}-\rho W^{\prime }\right)}^{-1}K & Q+\tau X^{\prime }X \end{array}\right)\\ & = & \left(\begin{array}{cc} \tau \left(I_{n}-\rho W^{\prime }\right)\left(I_{n}-\rho W\right) & -\tau \left(I_{n}-\rho W^{\prime }\right)\left(I_{n}-\rho W\right){\left(I_{n}-\rho W\right)}^{-1}X\\ -X^{\prime {\left(I_{n}-\rho W^{\prime }\right)}^{-1}}\tau \left(I_{n}-\rho W^{\prime }\right)\left(I_{n}-\rho W\right) & Q+\tau X^{\prime }X \end{array}\right)\\ & = & \left(\begin{array}{cc} \tau \left(I_{n}-\rho W^{\prime }\right)\left(I_{n}-\rho W\right) & -\tau \left(I_{n}-\rho W^{\prime }\right)X\\ -\tau X^{\prime \left(I_{n}-\rho W\right)} & Q+\tau X^{\prime }X \end{array}\right). \end{array}$$

This shows that for the given hyperparameters τ and ρ, the mean and precision matrix of (x,β) are 0 and P; that is, the constructed latent class x has a GMRF structure with the mean 0 and precision matrix P, which is consistent with the INLA framework, and thus can be implemented with the help of MCMCINLA.

#### 3.1.2. GMRF Structure of the Imputation Model

For the covariates $X=\left(X_{mis},X_{obs}\right)$ containing missing data, $X_{mis}$ denotes the part with the missing values and $X_{obs}$ denotes the part that is observable. We define the imputation submodel of the covariates as the latent effect $x^{\prime }=\left(x_{mis}^{\prime },x_{obs}^{\prime }\right)$, and set different priors for $x_{mis}^{\prime }$ and $x_{obs}^{\prime }$, respectively, where the observed term $x_{obs}^{\prime }$ in the latent effect is set to the mean equal to $X_{obs}$ and a high precision matrix (e.g., here, we take it as $5^{10}I$) so that its variance is very small, i.e., it makes the observed term $x_{obs}^{\prime }$ in the latent effect as infinitely close to the observed covariate data $X_{obs}$ as possible [19]. A spatial model with the mean $\mu_{c}$ and precision matrix $Q_{c}$ is built for the missing term $x_{mis}^{\prime }$ in the latent effect to impute the missing covariate data, and the procedure is as shown below.

The imputation model, that is, given the observed data $X_{obs}$ and the hyperparameter θ, provides the distribution of the missing values $X_{mis}$; hence, we have:(6)$$\pi \left(X_{mis}|X_{obs}\right)=\int \pi (X_{mis},\theta |X_{obs})d\theta =\int \pi (X_{mis}|X_{obs},\theta )\pi (\theta |X_{obs})d\theta .$$

Since $\pi \left(\theta |X_{obs}\right)$ is only related to the observed data $X_{obs}$, it can be considered as a priori of θ, which can further rewrite $\pi \left(\theta |X_{obs}\right)$ as:(7)$$\pi \left(\theta |X_{obs}\right)\propto \pi \left(X_{obs}|\theta \right)\pi \left(\theta \right).$$

In the general case, we assume that the covariates $X=\left(X_{mis},X_{obs}\right)$ follow a multivariate normal distribution of:$$X|\theta \sim Normal\left(\left(\begin{array}{c} \mu_{mis}\\ \mu_{obs} \end{array}\right),\left(\begin{array}{cc} Q_{mis,mis} & Q_{mis,obs}\\ Q_{obs,mis} & Q_{obs,obs} \end{array}\right)\right),$$
and then it defines that its imputation model will obey:$$X_{mis}|X_{obs},\theta \sim Normal\left(\mu_{c},Q_{c}\right),$$
where $\mu_{c}=\mu_{mis}-Q_{mis,mis}^{-1}Q_{mis,obs}\left(X_{obs}-\mu_{obs}\right)\;\mathrm{and}\;Q_{c}=Q_{mis,mis}$.

Considering that the covariates in the SLM are spatially correlated, the missing data in the covariates are imputed using a conditional autoregressive specification (CAR) spatial model-based approach. Under CAR, the mean of the model is set as $\mu =\alpha^{T}$ and the precision matrix as Q=τ(I−ρW), where α is the intercept of the linear predictor, τ is the precision parameter, ρ is the spatial autocorrelation parameter, W denotes the adjacency matrix, and the hyperparameter θ at this time consists of τ, ρ, and α.

Substituting the values of μ and Q, the covariates X follow a multivariate normal distribution of:(8)$$X|\theta \sim Normal\left(\left(\begin{array}{c} \mu_{mis}\\ \mu_{obs} \end{array}\right),\left(\begin{array}{cc} Q_{mis,mis} & Q_{mis,obs}\\ Q_{obs,mis} & Q_{obs,obs} \end{array}\right)\right)\;=Normal\left(\left(\begin{array}{c} \alpha_{mis}^{T}\\ \alpha_{obs}^{T} \end{array}\right),\left(\begin{array}{cc} \tau \left(I_{mis}-\rho W_{mis,mis}\right) & -\tau \rho W_{mis,obs}\\ -\tau \rho W_{obs,mis} & \tau \left(I_{obs}-\rho W_{obs,obs}\right) \end{array}\right)\right),$$
and the imputation model based on the spatial model will obey:(9)$$X_{mis}|X_{obs},\theta \sim Normal\left(\mu_{c},Q_{c}\right),$$
where $\mu_{c}=\alpha_{mis}^{T}-{\left(I_{mis}-\rho W_{mis,mis}\right)}^{-1}\left(-\rho W_{mis,obs}\right)\left(X_{obs}^{T}-\alpha_{obs}^{T}\right)Q_{c}=\tau \left(I_{mis}-\rho W_{mis,mis}\right).$ The hyperparameters τ,ρ,and α are obtained from $\pi \left(\tau ,\rho ,\alpha |X_{obs}\right)$, which is proportional to $\pi \left(X_{obs}|\tau ,\rho ,\alpha \right)\pi \left(\tau ,\rho ,\alpha \right)$.

Thereby, the latent effect $x^{\prime }=\left(x_{mis}^{\prime },x_{obs}^{\prime }\right)$ will obey the following multivariate normal distribution:(10)$$x^{\prime }|\theta \sim Normal\left(\left(\begin{array}{c} \mu_{c}\\ X_{obs} \end{array}\right),\left(\begin{array}{cc} Q_{c} & 0\\ 0 & 5^{10}I \end{array}\right)\right),$$
where $\mu_{c}$ and $Q_{c}$ are taken as shown in Equation (9). This shows that the latent effect $x^{\prime }$ that we defined for the imputation model has a GMRF structure that can be applied to the INLA framework, and thus it can be implemented with the help of MCMCINLA.

### 3.2. Implementation of the MCMCINLA Algorithm

When using MCMCINLA to deal with the SLM latent model with random missing data in covariates, it is first necessary to define an imputation submodel to impute the covariates with missing values so as to substitute the complete covariate data back into the SLM, and then use MCMCINLA to fit the model for estimation. The core of the MCMCINLA algorithm parameter estimation lies in dividing the estimated parameters into two groups: the first group is estimated using the Metropolis–Hastings (MH) algorithm in MCMC and the second group is estimated using the Bayesian model averaging (BMA) algorithm [21] in INLA. Therefore, the whole algorithmic process is carried out in three main steps:
Imputation of the missing covariates $X=\left(X_{mis},X_{obs}\right)$ using INLAIn this paper, we chose to impute the missing covariates $X=\left(X_{mis},X_{obs}\right)$ using a CAR space model-based approach, which first requires the definition of the mean, precision, hyperparameters, and prior of each hyperparameter in the latent effect $x^{\prime }$. The key codes are as follows: *inla.rgeneric.micar.model = function(cmd = c(“graph”, “Q”, “mu”, “initial”,”log.norm.const”, “log.prior”, “quit”), theta = NULL),*which define the spatial weight matrix (“*graph*”), precision matrix (“*Q*”), mean (“*mu*”), hyperparameter prior (“*log.prior*”), etc. in the latent effect $x^{\prime }$ by the *inla.rgeneric.micar.model*, respectively, in preparation for defining the imputation model;
*model = inla.rgeneric.define(inla.rgeneric.micar.model, debug = TRUE, n, x, idx.mis, W),*which defines the imputation model via the *inla.rgeneric.define()*, where *n* denotes the total number of indices, *x* denotes the covariates containing the missing data, *idx.mis* denotes the index of each missing datum, and *W* is the spatial weight matrix; and *inla(x~0 + f(idx, model = model),data,…),*where, finally, INLA is used to complete the fit of the imputation model, and where *f(idx, model =model)* represents the spatial effect of the imputation model. The covariate X after imputation via INLA is incorporated into the SLM, at which time there are three parameters to be estimated in the model, namely, the spatial autocorrelation parameter ρ, the covariate coefficient β, and the error term precision τ. Here, we use MH estimation for the parameter ρ and BMA estimation for the remaining parameters.Estimation of the spatial autocorrelation parameter ρ using MHThe estimation of ρ using MH is carried out in three main steps, as follows:
Step 1: Assume that starting from the initial point $\rho^{\left(1\right)}=0$, the model is fitted conditionally with $\rho^{\left(1\right)}$ to obtain $\pi (Y|\rho^{\left(1\right)})$, $\pi (\beta |Y,\rho^{\left(1\right)})$, and $\pi (\tau |Y,\rho^{\left(1\right)})$;Step 2: Use the MH algorithm to sample from the posterior of ρ, propose a new point $\rho^{*}$ for ρ by proposing the distribution $q\left(\cdot |\rho^{\left(j-1\right)}\right)$, fit the model conditionally on $\rho^{*}$ to obtain $\pi \left(Y|\rho^{*}\right)$, $\pi \left(\beta |Y,\rho^{*}\right)$, and $\pi \left(\tau |Y,\rho^{*}\right)$, and calculate $\pi \left(\rho^{*}\right)$, $q\left(\rho^{*}|\rho^{\left(j\right)}\right)$, and $q(\rho^{\left(j\right)}|\rho^{*})$;Step 3: Calculate the acceptance probability α of $\rho^{*}$ and determine whether the proposal is acceptable (or not), where:(11)$$\alpha =\min\left\{1,\frac{\pi \left(\beta |Y,\rho^{*}\right)\pi \left(\tau |Y,\rho^{*}\right)\pi \left(Y|\rho^{*}\right)\pi \left(\rho^{*}\right)q\left(\rho^{\left(j\right)}|\rho^{*}\right)}{\pi \left(\beta |Y,\rho^{\left(j\right)}\right)\pi \left(\tau |Y,\rho^{\left(j\right)}\right)\pi \left(Y|\rho^{\left(j\right)}\right)\pi \left(\rho^{\left(j\right)}\right)q\left(\rho^{*}|\rho^{\left(j\right)}\right)}\right\}.$$

If the proposal is accepted, then $\rho^{\left(j+1\right)}=\rho^{*}$ with $\pi \left(\beta |Y,\rho^{\left(j+1\right)}\right)=\pi \left(\beta |Y,\rho^{*}\right)\;\mathrm{and}\;\pi \left(\tau |Y,\rho^{\left(j+1\right)}\right)=\pi \left(\tau |Y,\rho^{*}\right)$; otherwise, $\rho^{\left(j+1\right)}=\rho^{\left(j\right)}$, and $\pi \left(\beta |Y,\rho^{\left(j+1\right)}\right)=\pi \left(\beta |Y,\rho^{\left(j\right)}\right)$ and $\pi \left(\tau |Y,\rho^{\left(j+1\right)}\right)=\pi \left(\tau |Y,\rho^{\left(j\right)}\right)$. This iterative process is executed until the end of the estimation.

The key code for the process is as follows:
*fit.inla < −slm.inla (formula, d, W, rho,…).*
in which we fit the SLM by defining *fit.inla()* to prepare for subsequent sampling, where *formula* is a formula with the response variable and the fixed effects, *d* represents the complete data set after imputation, *W* is the spatial weight matrix as above, and *rho* is the spatial autocorrelation parameter; and *INLAMH (d.mis, fit.inla, d.init, rq, dq, prior, n.sim = 200, n.burnin = 100, n.thin = 1,…),*in which we fit the parameters ρ by *INLAMH()*, where *fit.inla* is the model fitted earlier, and here, namely, the SLM, and *d.init*, *rq*, *dq*, and *prior* are all basic settings in the MH algorithm which denote the initial value of sampling, the proposed distribution, the density function of the proposed distribution, and the prior distribution, respectively.
3.Estimation of the coefficient β and precision τ using BMAFor the parameter β and hyperparameter τ, the conditional margins $\pi \left(\beta |Y,\rho^{\left(j\right)}\right)$ and $\pi \left(\tau |Y,\rho^{\left(j\right)}\right)$ generated by the MH algorithm in each iteration can be obtained using BMA, and further, the posterior margins of β and τ are derived by integrating over ρ, namely:(12)$$\pi \left(\beta |Y\right)=\int \pi \left(\beta |\rho ,Y\right)\pi \left(\rho |Y\right)d\rho =\frac{1}{N}\overset{N}{\underset{j=1}{\sum }}\pi \left(\beta |Y,\rho^{\left(j\right)}\right)\;\mathrm{and}$$
(13)$$\pi \left(\tau |Y\right)=\int \pi \left(\tau |\rho ,Y\right)\pi \left(\rho |Y\right)d\rho =\frac{1}{N}\overset{N}{\underset{j=1}{\sum }}\pi \left(\tau |Y,\rho^{\left(j\right)}\right).$$

The key code for the process is as follows:
*INLABMA:::fitmatrixBMA(l.models, ws, “summary.fixed”)*
*INLABMA:::fitmatrixBMA(l.models, ws, “summary.hyperpar”)*
which calculate the posterior margins for fixed effects and hyperparameters by the *fitmatrixBMA()* function in the INLABMA package and where *l.models* are the INLA models to be averaged and *ws* is the weight vector.

## 4. Simulation Study

### 4.1. Data Generation

Assuming that $\left(I_{n}-\rho_{Lag}W\right)$ is invertible, we consider the numerical simulation process of the SLM latent model for covariates with random missing data as follows:(14)$$Y={\left(I_{n}-\rho_{Lag}W\right)}^{-1}\left(X_{1}\beta_{1}+X_{2}\beta_{2}+e\right)+\varepsilon ,$$
where Y is the response variable, $X_{1}$ and $X_{2}$ are the covariates, and $\beta_{1}$ and $\beta_{2}$ are the coefficients corresponding to $X_{1}$ and $X_{2}$, respectively, $I_{n}$ is an n×n unit matrix, W is an n×n spatial weight matrix, e is a random error term with $e\sim N\left(0,\sigma^{2}\right)$, and ε is a fitting error term.

The specific simulation data are taken as follows:
In the main model, $X_{1}\sim U\left(0,1\right)\;\mathrm{and}\;X_{2}\sim U\left(0,1\right),\;\beta_{1}=0.3\;\mathrm{and}\;\beta_{2}=0.5,\;\rho_{Lag}=0.9$, and *Y* is generated by Equation (14) and we randomly remove 15% data as the missing values in $X_{1}$ using the MAR mechanism;In the imputation model, α=0.5 and ρ=0.2; andThe spatial weight matrices W in both the main model and the imputation model are selected as Queen-type adjoints, created in the regular lattice, and all error terms in the model obey the normal distribution $N\left(0,0.5^{2}\right)$, taking the sample size n = 250, pre-burn simulation of 20 times, interval rejection after pre-burn to keep one of the 5 iterations, and, finally, a total of 80 iterations of simulation are completed.

### 4.2. Fitting Effect Evaluation Indicators

The mean square error (MSE) value is used to reflect the difference between estimates and estimates, the deviance information criterion (DIC) value is used to measure the fit of the Bayesian model, and the accuracy is used to measure the prediction of the missing data. The MSE can evaluate the degree of change of the data, and the smaller the value of the MSE, the higher the accuracy of the data; the DIC criterion can weigh the complexity of the estimated model and the goodness of the model fit, and the smaller the value, the better the model fit; the accuracy is the ratio of the predicted value to the true value, which is used to reflect the similarity between the true value and the predicted value of the missing data and capture the missing information, and the larger the value, the more accurate the prediction result.

### 4.3. Results Analysis

Using MCMCINLA to first impute the missing data in the covariates by the method based on the CAR spatial model, and then fit the estimates to the SLM latent model, the results of each simulation are shown in Table 1 and Table 2, the fitted curves of $\rho_{Lag}$, $\beta_{1}$, and $\beta_{2}$ and τ and the prediction comparison plots of the missing data are shown in Figure 1, Figure 2 and Figure 3.

According to the results in Table 1, it can be found that the mean of each parameter $\beta_{1}$, $\beta_{2}$, $\rho_{Lag}$, and τ estimated in the SLM latent model are very close to the true values and the MSE and the DIC are small, indicating that the parameter estimation and model fitting of this estimation method are good. In addition, Table 2 shows the imputation results for each missing datum in the simulation, where “Mean” represents the mean of the predicted value of the missing data, “95%CI” denotes the 95% credible interval of the predicted value of the missing data, “True value” represents the true value of the missing data, and “Accuracy” is the ratio of “Mean” to “True value”. As shown in Table 2, except for the prediction accuracy of V7 and V34, which is around 85%, the prediction accuracy of each missing datum is basically above 90%, indicating that the imputation accuracy of our method is high and the information contained in the missing data can be effectively mined for research with the help of this method.

Figure 1 and Figure 2 show the fitted curves of the covariate coefficients $\beta_{1}$ and $\beta_{2}$, the error term precision τ, and the density function curve of the spatial autocorrelation parameter $\rho_{Lag}$, respectively, where the black solid line is the fitted value and the black dashed line perpendicular to the x-axis is the true value. Figure 3 shows the predicted versus true values of the missing data, where the *x*-axis represents the predicted values of the missing data and the *y*-axis represents the true values of these missing data that are artificially removed. As can be seen from Figure 1 and Figure 2, except for the peak of $\rho_{Lag}$, which deviates slightly from the true value, the peaks of all the other parameters are very close to the true value, and all points in Figure 3 basically fall on the line y=x. This indicates that the SLM latent model with the covariates containing missing data can be estimated and predicted relatively well using MCMCINLA.

## 5. Empirical Analysis

### 5.1. Subject Presents

In the post-epidemic era, outbreaks of various infectious diseases, especially the outbreak of COVID-19, have raised thoughts about the restructuring of the public health system and the corresponding financial investment reforms [22]. How to effectively adjust to the various problems revealed under the epidemic to ensure the continued health of the country’s public health has become a key and urgent issue for current research. In the case of medical data, data are often missing during collection and transmission due to clinical trials or equipment failures [23]. It is important to make full use of the large amount of data to mine the important information and make correct predictions of the missing data to make a comprehensive analysis of the research subject.

This paper obtains the economic and disease data of 31 regions in mainland China from 2016 to 2018 through the China Statistical Yearbook (http://www.stats.gov.cn, accessed on 31 December 2020) and the Public Health Science Data Center (http://www.phsciencedata.cn, accessed on 31 December 2018), and uses the national financial investment in public health as the response variable Y and the number of infectious disease cases, economic development, and scientific and technological development as the covariates X to construct the SLM latent model to explore how public health in China can further be developed in the post-epidemic era. The national financial investment in public health is expressed by the indicator “health expenditure in general public budget expenditure by region”. Since China has become the most seriously affected country in the world by HFRS [24], the infectious disease studied here is HFRS, and the number of infectious disease cases is expressed by the indicator “number of HFRS cases by region”. The economic development is expressed by the indicator “regional gross product”. The scientific and technological development is expressed by the indicator “number of research and experimental development R&D projects by region”. Using the MAR mechanism to randomly remove 15% of the data from the number of infectious disease cases to construct the SLM latent model for covariates with random missing data, we have:(15)$$Y={\left(I_{n}-\rho_{Lag}W\right)}^{-1}\left(X_{1}\beta_{1}+X_{2}\beta_{2}+X_{3}\beta_{3}+e\right)+\varepsilon .$$

If we set the prior distribution for the hyperparameters, the $\rho_{Lag}$ of the main model is assigned a uniform distribution that satisfies (−2, 1), and at this time, the minimum eigenvalue of the adjacency matrix W is −0.5. The prior of the coefficient vector $\beta_{i}$ and τ are set as the default values in R-INLA. Then, we assign a Gaussian prior with a zero mean and 0.00005 precision to the intercept α in the imputation model, set in the imputation model as above using the default settings in R-INLA, and assign logit(ρ) to a Gaussian prior with a zero mean and 0.001 precision.

### 5.2. Exploring the Development of Public Health in China in the Post-Epidemic Era

Using MCMCINLA, the missing data are imputed with the help of *inla.rgeneric.define()*, and the model is fitted using *fit.inla*( ) and the estimated values of the spatial autocorrelation parameters and the posterior estimates of the regression coefficients of the influencing factors are obtained as shown in Table 3. The regression coefficients of the influencing factors are plotted in Figure 4.

As estimated by MCMCINLA, the spatial autocorrelation parameter is $\rho_{Lag}$ = 0.6933, which indicates that there is a significant spatial correlation between the national financial investment in public health between regions. To a certain extent, the public health development of a region also has some influence on the public health development of the surrounding regions. From the results of Table 3, it can be found that the posterior mean of economic development is 1.2719, which indicates that the increase of economic level will increase the national financial investment in public health, which can provide the power financial source for the development of public health. However, the posterior mean of the number of infectious disease cases and scientific and technological development are −0.0421 and −0.3777, respectively, indicating that the incidence of infectious diseases and the country’s scientific and technological development during 2016–2018 have the opposite effect on the amount of financial investment in public health, which, to some extent, also reveals the shortcomings and inadequacies of the public health management system and the institutional mechanism for epidemic prevention and control. The increase in the number of incidences of general infectious diseases did not attract sufficient attention to the refinement of the CDC structural system, and the rising level of science and technology did not contribute to the problem of updating and replenishing the equipment of specialized public health institutions, leading to the sudden outbreak of the COVID-19 resulting in 2019 a shortage of medical resources and medical personnel and insufficient reserves of premises and materials [25]. Therefore, in the post-epidemic era, it is more important for the government to learn from previous experiences and lessons, establish a reserve mechanism of materials for public health emergencies, build a modern epidemic prevention material reserve system [26], increase the construction of professional public health institutions and the procurement of professional equipment updates, and improve the ability to face public health emergencies in order to quickly and effectively control sudden major infectious epidemics.

### 5.3. Imputation of Missing Predictor Values

Using the CAR spatial model-based approach to impute the SLM latent model with the missing data in $X_{1}$, we obtain the prediction information of the missing data and comparing the imputed predicted value with the true value to determine the imputation accuracy.

Based on the prediction results of the missing data in Table 4, it can be seen that the predicted means of the missing values are very close to the true values, and the accuracy rates are basically above 90%, indicating that the imputation of the missing covariates using the CAR space model-based approach with the help of MCMCINLA works well. Meanwhile, observing the data information in the table, we can find that the number of HFRS cases in different regions varies significantly, with a certain spatial heterogeneity and a spatial distribution trend of south-heavy and north-light. The areas with more incidences are Guangdong and Fujian, mostly in the mountainous areas in the south, which have richer vegetation and more precipitation; the areas with fewer incidences are Xinjiang, Tibet, Ningxia, and Gansu, which are mostly in the drier plains. To some extent, increased precipitation promotes the growth of vegetation and crops and provides a more suitable environment for rodents, such as rats, in mountainous areas, creating an increased risk of HFRS transmission [27]. Therefore, for HFRS, the relevant CDC departments should focus on the prevention and control of the epidemic in the southern region in the future to effectively control the further spread of the epidemic.

### 5.4. Comparison of Different Imputation Methods

With the development of statistical techniques, a series of model-based missing data processing methods, such as the maximum likelihood estimation, have received increasing attention from academics [28]. They mainly include EM and FIML, which have the advantages of convenient operation and more applicable models.

EM is an iterative algorithm that processes the missing data by calculating the maximum likelihood, and its imputation of the missing values can be achieved by continuous iteration once the initial values of the estimated parameters are given [29]. The principle of FIML is to model the available data using a “one-step” operation and estimate the parameters using a likelihood function; thus, the missing data imputation and parameter estimation processes are implemented simultaneously [30]. The missing data imputation process of the EM and FIML methods is done with the help of the TestDataImputation package and GDINA packages, respectively. The EM and FIML are imputed separately for the missing infectious disease case data and the results are obtained as shown in Table 5 and Table 6. The comparison shows that all three imputation methods can obtain more accurate estimates, and the results obtained by the different methods do not differ much. Relatively speaking, the imputation accuracy of MCMCINLA is slightly better than EM, and the FIML method ranks last; however, in terms of computational speed, both MCMCINLA and EM need to perform imputation before completing the estimation, and the estimation of ρ when MCMCINLA needs to be performed with the help of MH sampling, which takes a long time, takes about 0.5 h, while the FIML method can obtain both the imputed and estimated values at the same time, which is more efficient. In addition to using the accuracy to evaluate the imputation effect of each missing datum from the individual point of view, we can also select three evaluation indicators: MSE, mean absolute percentage error (MAPE) and Pearson correlation coefficient (r) to compare the imputation performance of the three methods from the global point of view [31]. The results are shown in Table 7. MSE can be used to measure the absolute deviation between the imputed value of the missing data and the true value, MAPE can be used to measure the relative error between the imputed value and the true value, both of which are as small as possible, and r can be used to illustrate the correlation between the imputed value and the true value, and the larger the r value, the better the fitting effect. From the results in Table 7, it can be found that the MSE and MAPE values of the three algorithms of MCMCINLA, EM, and FIML are all small, and the r values are all large, indicating that the three algorithms can obtain ideal imputation effects for missing data, and in comparison, the imputation performance of the MCMCINLA proposed in this paper is more prominent.

## 6. Conclusions and Discussion

Medical data are often missing during collection and transmission due to clinical trials or equipment failures. In this paper, we investigate the problem of parameter estimation for SLM when the covariates contain random missing data, and we propose a new imputation method that uses MCMCINLA to get not only accurate parameter estimates for the model, but also good imputation results for the missing data. Taking the economic and HFRS disease data of mainland China from 2016–2018 as an example for empirical analysis, the study found that the HFRS epidemic in China had obvious spatial heterogeneity and a south-heavy and north-light distribution trend. Before the outbreak of COVID-19 in 2019, China’s public health management system had certain problems, and the state’s financial investment in public health did not receive certain attention. Compared with EM and FIML, the predicted values of the missing data obtained using our method are closer to the true values. Therefore, in the future, the relevant CDC departments should focus their attention on the south or areas with a high incidence of epidemics in wetter climatic conditions and do a good job of the research and diagnosis of HFRS epidemics. In the post-epidemic era, the government should play a leading role, actively learn from previous experiences and lessons, establish a mechanism for stockpiling materials for public health emergencies, build a modern epidemic prevention material reserve system, and increase the construction of professional public health institutions and the procurement of professional equipment updates in order to quickly and effectively control sudden major infectious epidemics.

Since this paper mainly focuses on model imputation, fitting, and estimation with random missing data, the other two mechanisms are not explored in depth. It can continue to be extended in the future to study how to use the method to deal with different imputation models and missing mechanisms.

One of the fundamental aspects of imputation, in addition to the missing pattern (MAR, MCAR, and NMAR), is the percentage of missing data. It would be more intriguing to study how the new algorithm is affected by the percentage of missing data and compare its performance with other algorithms. In this paper, we take the 15% missing data percentage as an example to conduct intensive research. In the future, different levels of missing rates (such as 5%, 10%, 15%, 30%, and 50%) can be set to discuss the performance of the algorithm more comprehensively.

Finally, since in this paper we only design a set of simulation experiments to test the algorithm, objectively speaking, the effectiveness of the algorithm is questionable when it is extended to other situations (for example, if X takes different probability distributions, $\rho_{Lag}$ takes different spatial autocorrelation degrees). Therefore, the test effect of the imputation algorithm in other cases can be further explored in the future.

## Figures and Tables

**Figure 1 entropy-24-00916-f001:**
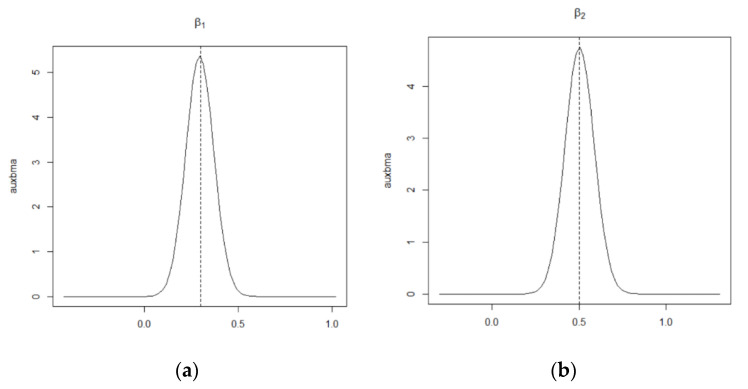
The fitted plots of $\beta_{1}$ and $\beta_{2}$ in the simulation. (**a**) Posterior density plot for the regression coefficient $\beta_{1}$ in the simulation. (**b**) Posterior density plot for the regression coefficient $\beta_{2}$ in the simulation.

**Figure 2 entropy-24-00916-f002:**
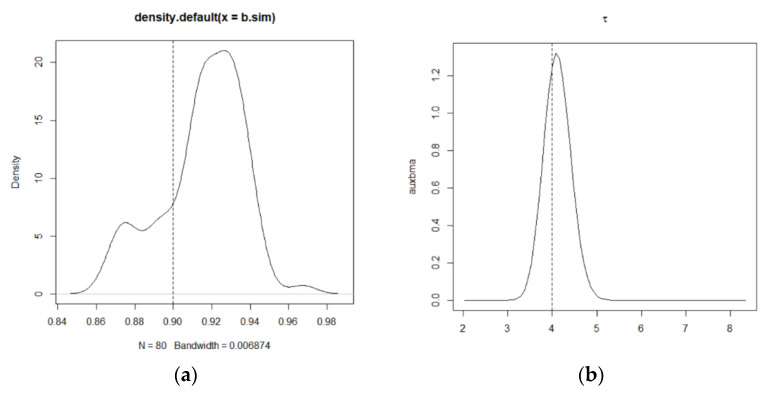
Density functional plot of $\rho_{Lag}$ and the fitted plot of τ in the simulation. (**a**) Density functional plot of the spatial autocorrelation parameter $\rho_{Lag}$ in the simulation. (**b**) Posterior density plot for the error term precision τ in the simulation.

**Figure 3 entropy-24-00916-f003:**
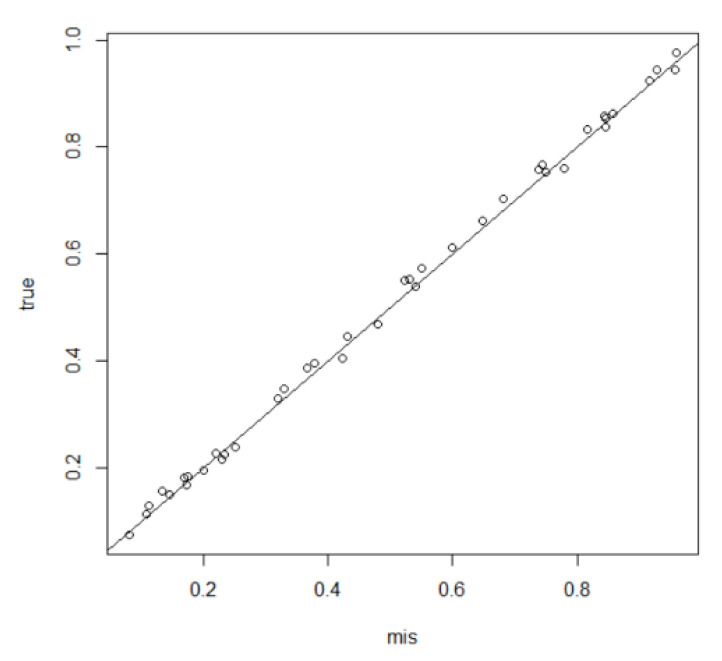
Predicted versus true values for the missing data in the simulation.

**Figure 4 entropy-24-00916-f004:**
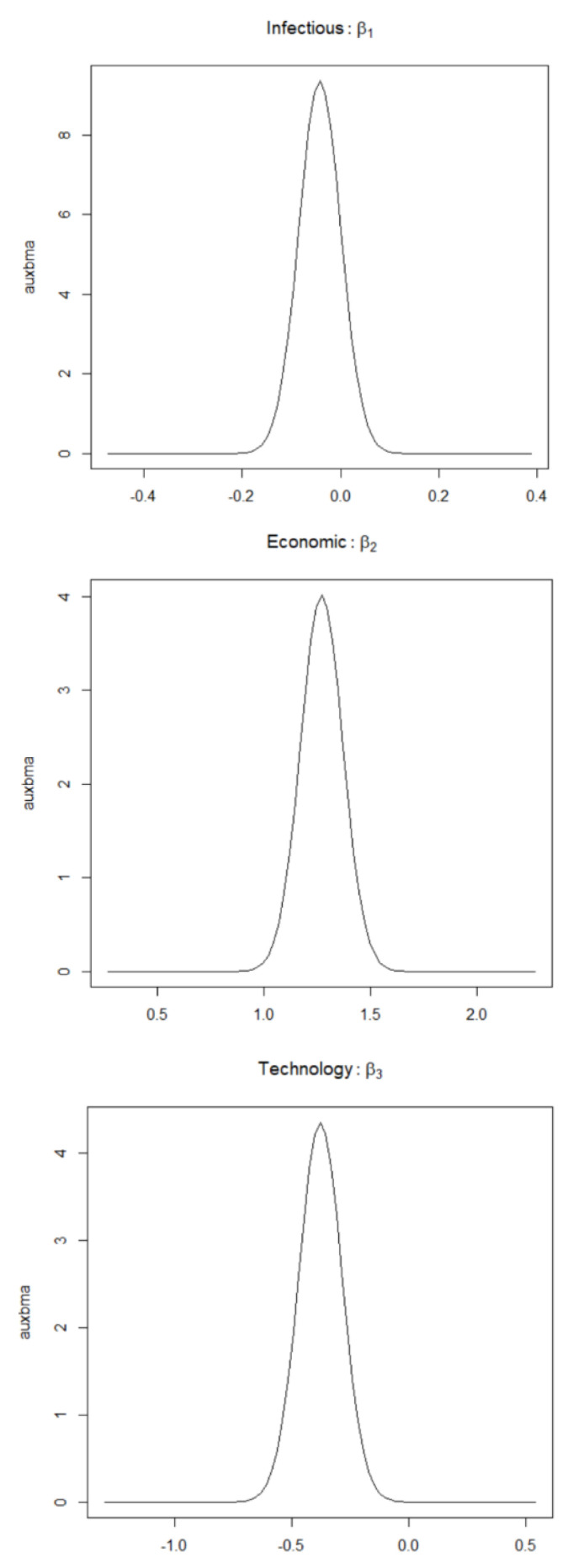
Estimated figures of the regression coefficients of each influencing factor of financial investment in public health.

**Table 1 entropy-24-00916-t001:** Parameter estimation results and model fitting results in the simulation.

	$\beta_{1}$	$\beta_{2}$	$\rho_{Lag}$	τ
Mean	0.2970	0.5040	0.9153	4.1133
Standard Deviation	0.0744	0.0840	0.0210	0.3036
95% Credible interval	(0.2463, 0.3466)	(0.4468, 0.5599)	(0.8711, 0.9432)	(3.9036, 4.3127)
MSE	2.25 × 10^−8^	4 × 10^−8^	5.929 × 10^−7^	3.209 × 10^−5^
DIC	12.7841			

**Table 2 entropy-24-00916-t002:** Imputation results for each missing datum in the simulation.

	Mean	95%CI	True Value	Accuracy
**V1**	0.1732	(0.1010, 0.2454)	0.1848	0.9372
**V2**	0.6821	(0.6502, 0.7140)	0.7023	0.9712
**V3**	0.5500	(0.4928, 0.6073)	0.5733	0.9593
**V4**	0.1720	(0.1405, 0.2033)	0.1680	0.9791
**V5**	0.9578	(0.9284, 0.9872)	0.9438	0.9853
**V6**	0.9276	(0.9020, 0.9534)	0.9434	0.9832
**V7**	0.1101	(0.0118, 0.2084)	0.1291	0.8528
**V8**	0.8169	(0.7875, 0.8461)	0.8334	0.9802
**V9**	0.4790	(0.4479, 0.5105)	0.4680	0.9770
**V10**	0.5222	(0.4633, 0.5813)	0.5499	0.9496
**V11**	0.5312	(0.4862, 0.5763)	0.5526	0.9612
**V12**	0.2510	(0.1945, 0.3055)	0.2388	0.9513
**V13**	0.7798	(0.7504, 0.8092)	0.7605	0.9752
**V14**	0.1673	(0.0911, 0.2435)	0.1808	0.9253
**V15**	0.4223	(0.3773, 0.4673)	0.4052	0.9595
**V16**	0.8467	(0.8292, 0.8644)	0.8535	0.9920
**V17**	0.9601	(0.9390, 0.9810)	0.9763	0.9834
**V18**	0.2333	(0.2039,0.2627)	0.2258	0.9678
**V19**	0.4298	(0.4006,0.4594)	0.4448	0.9662
**V20**	0.0800	(0.0075,0.1558)	0.0749	0.9362
**V21**	0.6472	(0.6215, 0.6729)	0.6618	0.9779
**V22**	0.3664	(0.3075, 0.4253)	0.3875	0.9455
**V23**	0.8468	(0.8269, 0.8667)	0.8368	0.9881
**V24**	0.1436	(0.0908, 0.1967)	0.1505	0.9541
**V25**	0.3280	(0.2684, 0.3872)	0.3472	0.9447
**V26**	0.5998	(0.5814, 0.6186)	0.6114	0.9810
**V27**	0.3776	(0.3254, 0.4230)	0.3949	0.9561
**V28**	0.5401	(0.5300, 0.5500)	0.5396	0.9990
**V29**	0.8579	(0.8455, 0.8703)	0.8616	0.9957
**V30**	0.7492	(0.7491, 0.7618)	0.7524	0.9957
**V31**	0.7376	(0.7083, 0.7695)	0.7582	0.9728
**V32**	0.9165	(0.9066, 0.9264)	0.9238	0.9920
**V33**	0.3179	(0.2753, 0.3605)	0.3289	0.9665
**V34**	0.1323	(0.0294, 0.2355)	0.1563	0.8464
**V35**	0.2290	(0.1604, 0.2976)	0.2151	0.9393
**V36**	0.1075	(0.0586, 0.1564)	0.1128	0.9530
**V37**	0.2001	(0.1712, 0.2232)	0.1957	0.9780
**V38**	0.8436	(0.8235, 0.8640)	0.8576	0.9836
**V39**	0.2179	(0.1712, 0.2646)	0.2277	0.9569
**V40**	0.7447	(0.7214, 0.7683)	0.7670	0.9709

**Table 3 entropy-24-00916-t003:** Posterior estimates of the regression coefficients of each influencing factor of financial investment in public health.

Variables	Mean	Standard Deviation	95% Credible Interval
The number of infectious disease cases	−0.0421	0.0429	(−0.0711, −0.0135)
Economic development	1.2719	0.1001	(1.2043, 1.3385)
Scientific and technological development	−0.3777	0.0925	(−0.4399, −0.3160)

**Table 4 entropy-24-00916-t004:** Missing data prediction results in the number of infectious disease cases (MCMCINLA).

	Mean	95%CI	True Value	Accuracy
**Ningxia (2016)**	4.3678	(2.8478, 5.8889)	4	0.9157
**Tibet (2016)**	1.1752	(0.1642, 2.1894)	1	0.8509
**Hubei (2016)**	248.4683	(226.5048, 270.4318)	236	0.9498
**Guangdong (2016)**	436.1809	(391.1957, 481.1663)	410	0.9399
**Gansu (2016)**	20.4501	(11.3075, 29.5927)	19	0.9290
**Hebei (2016)**	456.7421	(406.4227, 510.0615)	434	0.9502
**Guangxi (2017)**	14.9736	(9.7006, 20.2477)	14	0.9349
**Beijing (2017)**	7.7038	(4.8404, 10.5663)	7	0.9086
**Xinjiang (2018)**	1.1431	(0.0471, 2.2392)	1	0.8748
**Shanxi (2018)**	22.5906	(14.3455, 30.8357)	21	0.9295
**Tibet (2018)**	1.1620	(0.1601, 2.1639)	1	0.8605
**Shandong (2018)**	1268.9431	(1025.2239, 1512.6623)	1218	0.9598
**Fujian (2018)**	447.8379	(408.5648, 487.1110)	430	0.9601
**Tianjin (2018)**	21.0547	(12.3284, 29.7810)	20	0.9499
**Chongqing (2018)**	13.0454	(7.7685, 18.3223)	12	0.9198

**Table 5 entropy-24-00916-t005:** Missing data prediction results in the number of infectious disease cases (EM).

	Mean	95%CI	True Value	Accuracy
**Ningxia (2016)**	4.3922	(2.8294, 5.9552)	4	0.9107
**Tibet (2016)**	1.1610	(0.1573, 2.1647)	1	0.8613
**Hubei (2016)**	255.4665	(212.1427, 298.7903)	236	0.9238
**Guangdong (2016)**	444.9267	(372.6629, 517.1905)	410	0.9215
**Gansu (2016)**	20.6387	(10.9749, 30.3025)	19	0.9206
**Hebei (2016)**	457.8542	(401.1974, 514.5110)	434	0.9479
**Guangxi (2017)**	15.3475	(9.0223, 21.6727)	14	0.9122
**Beijing (2017)**	7.9113	(4.1923, 11.6303)	7	0.8848
**Xinjiang (2018)**	1.1326	(0.0012, 2.2640)	1	0.8829
**Shanxi (2018)**	22.9332	(13.5426, 32.3238)	21	0.9157
**Tibet (2018)**	1.1372	(0.1477, 2.1266)	1	0.8793
**Shandong (2018)**	1292.1705	(1009.3498, 1574.9912)	1218	0.9426
**Fujian (2018)**	453.3473	(396.4419, 510.2527)	430	0.9485
**Tianjin (2018)**	21.4799	(11.3123, 31.6475)	20	0.9311
**Chongqing (2018)**	13.3288	(7.0392, 19.6184)	12	0.9003

**Table 6 entropy-24-00916-t006:** Missing data prediction results in the number of infectious disease cases (FIML).

	Mean	95%CI	True Value	Accuracy
**Ningxia (2016)**	4.4400	(2.7943, 6.0857)	4	0.9009
**Tibet (2016)**	1.1219	(0.1621, 2.0817)	1	0.8913
**Hubei (2016)**	256.9686	(210.1379, 303.7993)	236	0.9184
**Guangdong (2016)**	450.1042	(368.4492, 531.7592)	410	0.9109
**Gansu (2016)**	20.9320	(10.8787, 30.9853)	19	0.9077
**Hebei (2016)**	468.7837	(392.1163, 545.4583)	434	0.9258
**Guangxi (2017)**	15.5936	(8.9372, 22.2501)	14	0.8978
**Beijing (2017)**	7.8431	(4.3958, 11.2904)	7	0.8925
**Xinjiang (2018)**	1.1282	(0.0297, 2.2267)	1	0.8863
**Shanxi (2018)**	23.2970	(12.4424, 34.1517)	21	0.9014
**Tibet (2018)**	1.1462	(0.1392, 2.1532)	1	0.8724
**Shandong (2018)**	1312.6414	(1004.2167, 1621.0661)	1218	0.9279
**Fujian (2018)**	457.3981	(391.2553, 523.5409)	430	0.9401
**Tianjin (2018)**	21.7936	(11.0744, 32.5128)	20	0.9177
**Chongqing (2018)**	13.5379	(7.0102, 20.0656)	12	0.8864

**Table 7 entropy-24-00916-t007:** Imputation performance comparison of MCMCINLA, EM, and FIML.

Evaluation Indicators	MCMCINLA	EM	FIML
MSE	4.27 × 10^−5^	9.44 × 10^−5^	2.153 × 10^−4^
MAPE	3.498 × 10^−5^	7.2 × 10^−5^	1.058 × 10^−4^
r	0.9222	0.9122	0.9051

## Data Availability

All the data used in this paper can be obtained from the China Statistical Yearbook (http://www.stats.gov.cn, accessed on 31 December 2020) and the Public Health Science Data Center (http://www.phsciencedata.cn, accessed on 31 December 2018).

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
