# Peer review of "MCMCINLA Estimation of Missing Data and Its Application to Public Health Development in China in the Post-Epidemic Era"

_entropy, 2022, doi:10.3390/e24070916_

Round 1
Reviewer 1 Report
The paper is of interest for data scientists. It would be good to say something about the importance of the discussed topic for other data than medical ones.
Some more detailed descriptions of the term "precision" and "precision matrix" will be helpful.
...precision 5^10 , so that the ?_???
term is as close as possible to ?_???... what does it mean?
5^10?
Author Response
请参见附件。

Reviewer 2 Report
Overview:
The manuscript presents a new algorithm for data imputation, which is based on Markov Chain Monte Carlo (MCMC) and Integrated Nested Laplace Approximation (INLA). The authors describe the algorithm and present results for both a simulation and a real data problem. Imputation is a relevant issue with implications in several areas, particularly involving big data where the likelihood of missing data is high. The paper is well written, but in some parts, it is difficult to follow and could be improved.
Comments/suggestions:
Between lines 122 and 123, it is introduced the equation of the model SLM. This equation is not numbered and the y parameter is not identified, perhaps because there is a typo and it should be Y (capital letter). Please, clarify.
Pseudo-code written on lines 206-207, 209-210, 212-213, 232-233, 233-234, and 243-244 are difficult to understand since functions are not clearly defined. Please, clarify the pseudo-code or replace it with suitable flowcharts.
In lines 252 to 261, the “(…) specific simulation data” is defined, in the sense that the rationale for its creation is described. Using this specific simulation, the imputation algorithm was then tested. Thus, it seems to me that the generalization of the imputation algorithm to other situations (e.g., different parameters and different probability distributions for X) is debatable as only one situation was tested. My suggestion is to address this aspect in the discussion as a possible limitation.
The authors chose to present the mean of the imputed values (Tab. 2 for simulated data and Tab. 4, 5 and 6 for real data) together with the true value and the corresponding accuracy. The accuracy is defined (line 269) as “the ratio of the predicted value to the true value”. It is not clear whether the accuracy presented in the tables is the mean value of the accuracies or if it was obtain from the mean of the predicted values. Please, clarify. Furthermore, as the extreme situations of algorithm failure are the most interesting to be explored, at least one measure of dispersion of the accuracy or the predicted value should be presented. My suggestion is to show the 2.5th and 97.5th percentiles.
In addition to individual accuracies, a global measure of performance should be included for two reasons: with a global measure it is easier to compare the different imputations algorithms tested on real data, and individual measures can make a fair evaluation difficult, as the the average of the values can not be representative of particular compositions of the imputed values.
One of the fundamental aspects of imputation, in addition to the missing pattern (MAR, MCAR, and NMAR), is the percentage of missing values, which was not addressed in the work. It would be interesting to investigate how the novel algorithm is influenced by the percentage of missing data and to compare its performance with the other algorithms. I understand that including this analysis is very difficult due to the high effort required, but at least the authors can add a comment about it in the discussion. The authors mention (line 332) that 15 data (points?) were randomly removed, but it is unclear what percentage of missing values that “15” represents. Please, clarify.
How long does the algorithm take? Please include this information in the manuscript.
Do the authors intend to make the algorithm available for free?
